# Thin Layers of Cerium Oxynitride Deposited via RF Sputtering

**DOI:** 10.3390/ma17133142

**Published:** 2024-06-27

**Authors:** Gloria Carolina Numpaque, Manuel Bethencourt, Gloria Ivonne Cubillos

**Affiliations:** 1Grupo de Materiales y Procesos Químicos, Departamento de Química, Universidad Nacional de Colombia, Av. Cra. 30 No 45-03, Bogotá 16486, Colombia; gnumpaquer@unal.edu.co (G.C.N.); gcubillos@unal.edu.co (G.I.C.); 2Department of Materials Science, Metallurgical Engineering and Inorganic Chemistry, Institute of Marine Science (INMAR), CASEM, Polígono del Rio San Pedro s/n, Puerto Real, University of Cadiz, 11510 Cadiz, Spain

**Keywords:** cerium oxynitride, RF sputtering, co-sputtering, thin films, coatings, corrosion

## Abstract

Thin films of transition metal oxides and oxynitrides have proven highly effective in protecting stainless steels against corrosion in both chemically aggressive environments and biological fluids. In the present work, cerium zirconium oxynitride thin films were deposited to enhance the corrosion resistance of surgical-grade stainless steel to be used in osteosynthesis processes. Two techniques were employed: co-sputtering and radiofrequency (RF) sputtering, and the morphology and corrosion efficiency of the coatings deposited by each technique were evaluated. X-ray diffraction, X-ray photoelectron spectroscopy and field emission transmission electron microscopy were used to characterize the morphological and chemical structure, respectively. Additionally, the corrosion resistance of the oxynitride-coated surgical grade stainless steel system (ZrCeO_x_N_y_-AISI 316L) was assessed using Hank’s solution as the corrosive electrolyte, to determine its resistance to corrosion in biological media. The results show that ZrCeO_x_N_y_ coatings increase the corrosion resistance of surgical grade stainless steel by two orders of magnitude and that the Ce(III)/Ce(IV) equilibrium decreases the corrosion rate, thereby increasing the durability of the steel in a biological environment. The results show that Ce coatings increase the corrosion resistance of surgical grade stainless steel by two orders of magnitude and that the Ce(III)/Ce(IV) equilibrium decreases the corrosion rate, thereby increasing the durability of the steel in a biological environment.

## 1. Introduction

The permanent technological challenge in engineering and medicine requires the development and production of new materials with superior properties to meet emerging demands. The technological viability of a material depends on its response when in service, considering mechanical, thermal, optical, refractory, and corrosion resistance properties. While a versatile material with properties applicable in different fields is sought, a single material rarely exhibits an ideal combination of these properties. In some cases, surface modification is employed to meet specific needs and applications. Particularly, thin film coatings are a commonly applied technique to modify the surface to achieve desirable properties without altering the underlying substrate [1,2]. In surface modification, the field of thin films has gained wide interest as a method to modulate material properties cost-effectively compared to bulk materials. For ceramics, materials with high chemical inertness, mainly oxides, nitrides, or carbides of titanium, zirconium, and lanthanide elements (such as cerium [2,3,4]), are sought. These substances are abundant in nature and have the ability to reverse oxidation/reduction without altering the lattice structure, given the redox characteristics of the Ce(III)/Ce(IV) pair [5]. This property of cerium, to reversibly transform its Ce(III) and Ce(IV) oxidation states, is fundamental for its functionality and potential use in the development of new materials.

Cerium oxide is known for its wide range of applications, including catalysis, corrosion prevention, electrochemical cells, photocatalysis, UV absorbers, biomaterials, microelectronics, optical devices, thermal coatings, and abrasives [1,2,6]. In three-way catalysts, mixed with ZrO_2_, it efficiently oxidizes CO, nitrogen oxides (NO_x_), and hydrocarbons [7], surpassing the traditional Pt/Al_2_O_3_ catalyst used in pollution control systems generated by automobiles [8,9]. CeO_2_, when associated with yttria-stabilized zirconia (YSZ), improves thermal cycle life by increasing corrosion and thermal shock resistance for thermal barrier coatings (TBCs). YSZ undergoes tetragonal to monoclinic phase transformations with a 6.5% unit cell volume increase in the temperature range of 1200 to 1500 °C, resulting in the fracture and delamination of the ceramic coating. The incorporation of CeO_2_ generates a higher thermal gradient in the TBC, maintaining lower temperatures in the lower layers than at the surface, thereby stabilizing the crystalline structure of the coating [10,11].

In aqueous solutions, cerium exhibits a large reduction potential of approximately 2.3 eV vs. NHE [12]. The redox potential of the Ce(IV)/Ce(III) pair is approximately +1.7 eV, classifying Ce(III) as a strong reducing agent and Ce(IV) as a potent oxidizing agent. Recent studies have indicated that cerium atoms in CeO_2_ must exist in the Ce^4+^ oxidation state. However, the presence of highly reducible crystal lattice oxygen vacancies implies the existence of Ce^3+^ ions near each vacancy, highlighting its widespread application in fuel cells [13,14,15].

Furthermore, the incorporation of nitrogen into the crystalline structure of the oxide results in a substantial enhancement in strength, toughness, and elastic modulus, emphasizing the significance of oxynitrides [16,17]. Zirconium oxynitride stands out for its optical properties and electrical conductivity, adjustable according to the oxygen content. Consequently, its behavior ranges from a semiconductor in ZrN to an insulator in ZrO_2_ [18,19,20]. Zirconium exhibits a higher electronic affinity toward oxygen than nitrogen, forming an energetically more favorable Zr–O bond than a Zr–N bond. This allows the easy incorporation of oxygen into the nitride until a homogeneous Zr_2_N_2_O phase is achieved [18]. Additionally, cerium oxide is characterized by excellent physicochemical properties, including a high dielectric constant, high temperature resistance, and chemical inertness [21,22,23,24,25]. Therefore, the combination of these two transition metals leads to the production of catalysts with high selectivity for specific reactions [26,27,28,29] and fuel cells [14,15,30].

This research describes the method used to deposit ZrCeO_x_N_y_-type bimetallic oxynitride films on surgical-grade 316L stainless steel using co-sputtering and radiofrequency (RF) sputtering techniques. Their behavior against corrosion in Hank’s balanced salt solution (HBSS), simulating the corrosion process in a physiological environment, was evaluated to assess their potential use in metallic prosthesis coatings that inhibit corrosive processes, preventing toxic reactions and chronic allergies in the human body [31,32]. HBSS is a widely used buffer in cell culture for maintaining viable cells, as it helps maintain osmotic balance and a pH similar to normal physiological conditions. The solution also has a concentration of inorganic ions comparable to that found in a homeostatic cellular environment, making it suitable for corrosion studies of materials with potential for orthopedic implants. The synthesized thin films demonstrated stability and high resistance to corrosion.

## 2. Materials and Methods

### 2.1. Substrate Preparation and Deposition Conditions

Specimens of 316L steel with an area of 2.0 × 2.0 cm^2^ were polished with SiC of varying grain sizes up to 600 grains using a Knuth Rotor-3 (Struers, Copenhagen, Denmark) grinder at 150 rpm. This selection was made because, after testing grain sizes from 400 to 1200 grit, the best results for human osteoblast cell growth (hFOB cell line) on AISI 316L were obtained with the 600-grit polished steel [33]. Impurities were removed by washing with an anionic surfactant in an ultrasonic cleaner, followed by washing 3 times for 3 min in water and 3 min in isopropyl alcohol (IPA). Subsequently, specimens were dried in a stream of purified air.

Zirconium and cerium oxynitride (CeZrO_x_N_y_) thin films were deposited on surgical-grade 316L stainless steel using two different techniques:Co-sputtering: Employing INTERCOVAMEX H2 equipment (Episerve, Munich, Germany) with a zirconium target and a 2-inch diameter-, 5 mm thick-cerium target (99.95% purity, Stanford Advanced Materials). The target–substrate distance remained constant at 10 cm, deposition temperature was 200 °C, with a power applied to the target at 150 W. Ar and O_2_ flow rates were 20.0 sccm and 2.00 sccm, respectively, with a nitrogen flow rate varying between 15.0 sccm and 40.0 sccm. Deposition time was 45 min and the working pressure was maintained at 6.7 × 10^−1^ Pa.Radiofrequency (RF) sputtering: Utilizing Alcatel HS 2000 (Alcatel Vacuum Technology, Annecy, France) equipment with a 13.8 MHz RF source. The targets used were high-purity Ce and Zr (99.95%), with a 2-inch and 4-inch diameter, respectively, and 5 mm thickness (Stanford Advanced Materials). Ar and O_2_ fluxes were 20.0 sccm and 2.00 sccm, respectively, and N_2_ flux varied between 15.0 and 40.0 sccm. The substrate target distance was kept constant at 5 cm, and the substrate temperature at 200 °C. Deposition time was 45 min, with a power applied to the target of 150 W, and the working pressure was constant at 1.4 × 10^−1^ Pa. To ensure coating adhesion to the substrate, a NiZrO_2_ buffer layer was deposited for 15 min at a power of 250 W and an Ar flow rate of 20 sccm. The target configuration used is presented in Figure 1, and the deposition conditions for each method are summarized in Table 1.

The conditions tested are presented in Table 1. The conditions that showed the best adhesion and corrosion resistance of the films were selected for this work.

### 2.2. Characterization of the Coatings

Surface morphology was assessed using scanning electron microscopy on two microscopes. Firstly, a Hitachi SU 1510 in high vacuum mode at a voltage of 10 or 15 kV, coupled to a Bruker energy dispersive X-ray spectroscopy detector (EDX) with an accelerating voltage of 15 kV, was used. Secondly, an FEI Nova NanoSem 450 (FEI Company, Hillsboro, Oregon, USA), with high vacuum resolutions of 30 kV, 15 kV, 1 kV, and 100 V, and low vacuum resolution at 10 kV and 3 kV, also coupled to an EDX, was used. Coating growth was evaluated from the cross-section cut for films deposited on glass.

Thickness was determined using a DEKTAK 150 profilometer with a resolution of 0.056 μm and a maximum vertical resolution of 50 nm. Structural analysis was performed using X-ray diffraction (XRD) in a Panalytical X-Pert Pro diffractometer operating with Bragg–Brentano geometry, a Cu cathode, a Ni filter to absorb K*β* radiation, and obtaining a monochromatic radiation Kα *λ* = 0.15418 nm. The scanning range was from 10° to 80°, with a step size of 0.02° in continuous mode. The phase composition was evaluated against the ICDD PDF-2 database.

### 2.3. Corrosion Resistance

Potentiodynamic linear polarization (LP) measurements were conducted in three-electrode cells, using a VersaSTAT 4 potentiostat. A Ag/AgCl/KCl reference electrode (+197 mV/SHE, 25 °C) was employed as the reference electrode, with a graphite counter electrode. The exposed surface of the working electrode was 1 cm^2^. Prior to LP, the open circuit potential (OCP) was measured for 180 min for potential stabilization. LPs were carried out in Hank’s solution (Table 2) at pH 7.4 from −0.05 V to 0.250 V vs. the corrosion potential (E_corr_) at a scan rate of 0.167 mV/s. The E_corr_ and corrosion current density (J_corr_) were determined from the polarization curves using the Tafel extrapolation method. The corrosion current densities of the samples were calculated using the Stern–Geary equation [34], with ±10 mV around the open circuit potential to calculate polarization resistance (R_p_).

## 3. Results and Discussion

### 3.1. Co-Sputtering

A representative sample for ZrCeO_x_N_y_ thin films deposited by the co-sputtering technique is presented in Figure 2. In general, the films obtained under different deposition conditions were characterized by homogeneous chemical composition. Cerium, zirconium, oxygen, and nitrogen were uniformly distributed throughout the coatings, as revealed by EDX mapping. While oxygen is associated with both the coating and the alloying elements of the steel (Fe, Cr, and Ni), the morphology of the films deposited with different nitrogen fluxes using co-sputtering technique was characterized by low adhesion to the substrate, leading to delamination from the AISI 316L steel. In the upper-left corner, a cross-sectional cut of the same films is shown. It will be shown later that this is the columnar growth of the coating. The thickness of the films deposited using this technique ranges from 200 nm to 250 nm.

Figure 3 shows the diffractograms for the thin films deposited with nitrogen fluxes of 15.0 sccm and 20.0 sccm on two substrates: AISI 316L and glass. Because glass is amorphous, it only allows the diffraction planes of the film to be identified. The diffraction planes of the AISI 316L substrate are identified with (*). The diffraction planes for the coatings have been identified based on JCPDS chart 00-043-1002 since the XRD pattern for ZrCeO_x_N_y_ has not been reported in the literature and the experimental diffraction pattern fits this pattern. For the two nitrogen flows on the two substrates, it is observed that the film is characterized by being polycrystalline with preferential growth on the (220) plane for ϕN_2_ = 15.0 sccm and on the (200) plane for ϕN_2_ = 20.0 sccm.

The delamination of these films (Figure 4a,b), favoring the accumulation of the corrosive electrolyte, led to substrate attack. For the presented samples, the corrosion current density of the ZrCeO_x_N_y_-316L system was lower by one to two orders of magnitude compared to the bare steel, and the corrosion potential was higher by 0.3 V compared to the same substrate (see Figure 4c, Table 3). Regarding the passivation zone, comparing bare steel vs. AISI 316L-ZrCeO_x_N_y_ system, it was lower by about 38% for the sample deposited with a nitrogen flow of 15.0 sccm and by 46% for the one deposited with ϕN_2_ = 20.0 sccm. This behavior was directly associated with film delamination.

Similar results have been reported by Zhang [35], in the assessment of corrosion resistance for Al–Co–Ce amorphous alloys. In this case, the passive CeO_2_ film, despite widespread cracking, enhances the alloy’s corrosion resistance compared to the variant lacking cerium. Additionally, Castaño [13] and L.M. Calado [36] have observed the protective effect of ceria nanoparticles on AZ31 magnesium alloy. This protection arises from the adsorption of aggressive species such as Cl^−^. Conversely, M.F. Montemor [37] noted the corrosion inhibition performance of cerium salt on galvanized steel substrates. The addition of CeO_2_ nanoparticles reduces corrosive activity by forming a more stable and protective surface film, thereby amplifying anodic passivation.

The only way cerium could function as a sacrificial anode, preventing the oxidation of iron in the steel, is in the Ce° state (Equation (1)). However, XPS analysis indicates an absence of Ce° in the film. Moreover, the series of chemical reactions during the corrosion process in a neutrally aerated medium reveals that Ce^4+^, with a significantly higher reduction potential than Fe^2+^, cannot serve as a sacrificial anode for iron (Equations (2)–(4)). Furthermore, according to the redox potential, Ce^4+^ would be reduced to Ce^3+^ while the Fe in the steel will be oxidized to Fe^2+^:
  Ce^3+^ + 3e^−^ → Ce^o^      −2.33 V(1)
  Ce^4+^ + e^−^ → Ce^3+^     +1.61 V(2)
Fe^2+^ + 2e^−^ → Fe      −0.44 V(3)
4e^−^ + O_2_ + 2H_2_O → 4 OH^−^  +0.401 V(4)

Contrarily, XRD showed no evidence of crystalline ZrO_x_N_y_. However, both EDX and XPS showed evidence of Zr^4+^, suggesting that, given the smaller ionic radius of Zr^4+^ (0.84 A°) compared to Ce^4+^ (1.01 A°), Zr^4+^ integrates into the crystalline structure of CeZrO_x_N_y_, forming a solid solution. This phenomenon of solid solution formation between Ce^4+^ and Zr^4+^ has been extensively documented in CeO_2_–ZrO_2_ catalysts, solid-state fuel cells, and sensors [25,26,27,28,38,39]. The introduction of Zr^4+^ enhances the number of oxygen vacancies. Considering the electronic configuration of cerium ([Xe] 4f^1^5d^1^6s^2^), where its two most stable oxidation states are Ce^4+^ and Ce^3+^, these oxygen vacancies facilitate the equilibrium between Ce^4+^/Ce^3+^. Notably, at surface oxygen vacancy sites, Ce^4+^ ions undergo reduction to Ce^3+^.

Corrosion tests were conducted at a neutral pH, and in accordance with the Pourbaix diagram for Ce [40], complexes, namely Ce(OH)^2+^ and Ce(OH)_2_^1+^, are formed in the pH range between 6–8 and above −0.2 V [13,35,37]. Specifically, at neutral pH, the OH^−^ produced by the reduction of oxygen (Equation (4)) forms stable complexes with Ce(III), inhibiting the corrosion process (Equations (5) and (6)):Ce^3+^+ OH^−^ → Ce(OH)^2+^(5)
Ce^3+^+ OH^−^ → Ce(OH)_2_^1+^(6)

The formation of these Ce(III) complexes blocks paths for electrolyte uptake, hinders the progression of the corrosion process, and consequently, significantly enhances the corrosion resistance of bare steel in the presence of the ZrCeO_x_N_y_ film [24,39].

In summary:Zr^4+^ induces oxygen vacancies in the crystal structure of CeO_x_N_y_;Ce^4+^ undergoes reduction to Ce^3+^ within the oxygen vacancies;The formation of stable Ce(III) complexes with the OH^−^ of the medium inhibits the corrosion process.

### 3.2. RF Sputtering

#### 3.2.1. Chemical Composition

For the deposition of ZrCeO_x_N_y_ thin films using the RF sputtering technique, the target configuration presented in Figure 1a was employed. All the films deposited under different conditions with this target configuration exhibited high homogeneity in terms of chemical composition and morphology, without any delamination zones. In Figure 5, the elemental analysis for a selected ZrCeO_x_N_y_ film is presented.

Table 4a provides the reported chemical composition for the coating elements, inclusive of the substrate alloying elements (Fe, Cr, and Ni). In Table 4b, only the coating chemical elements Zr, Ce, and N_2_ have been normalized to 100%. Oxygen is not included because it remains constant under different deposition conditions and is not exclusively bound to Ce and Zr but also to the chemical elements constituting the substrate, forming oxides such as Fe_2_O_3_, Cr_2_O_3_, and NiO.

The highest percentage of nitrogen in the coating was achieved under deposition conditions using nitrogen flow rates of 15 and 25 sccm (Table 4b), while the lowest was observed for the film deposited with the highest nitrogen flow rate (40.0 sccm). In terms of metal composition, cerium is consistently found to be 3 to 4 times higher than zirconium. Depending on the nitrogen flow used for the deposition of ZrCeO_x_N_y_ films, the cerium percentage deposited exhibits an increasing trend with the nitrogen flow (except for ϕN_2_ = 35.0 sccm, where it decreases to 67% of Ce). This behavior is likely associated with the higher *momentum* of cerium compared to zirconium, given its 35%-higher atomic mass and the larger exposed area of cerium according to the target configuration used. This configuration favors Ce atoms reaching the substrate with high energies, resulting in a film richer in cerium.

Regarding the nitrogen chemical composition, it remains constant (approximately 5% N_2_) up to a nitrogen flux of 30.0 sccm. Beyond this point, it decreases as the nitrogen flux increases, indicating that the cerium oxynitride crystal structure has reached the maximum degree of oxygen substitution by nitrogen for this flux (ϕN_2_ = 20.0 sccm does not follow this behavior, %N_2_ is 2.9%).

Chemical analysis from XPS shows that the chemical composition is dependent on the deposition conditions for cerium and nitrogen. However, zirconium and oxygen exhibit no discernible changes corresponding to the nitrogen flux employed during film deposition. Figure 6 illustrates the characteristic high-resolution spectra for Ce(III) and Ce(IV), while Figure 7 presents the deconvolution for a 3d Ce sample where cerium shows its two characteristic oxidation states for Ce(III) and Ce(IV), O1s and N1s. Finally, Figure 8 displays the collection of high-resolution spectra for Ce3d, Zr3d, Zr3p, N1s, and O1s obtained for ZrCeO_x_N_y_ films deposited with nitrogen flow from 15.0 sccm to 40.0 sccm.

Figure 6 illustrates the various multiplet signals corresponding to the Ce(III) and Ce(IV) spectra. A comprehensive examination of the full Ce3d envelope for Ce(IV) reveals a core-level spectrum centered at approximately 882 eV. This spectrum exhibits six peaks, corresponding to three pairs of spin–orbit doublets at 882.1 eV, 889.1 eV, and 898.0 eV for Ce3d_5/2_, and 900.5 eV, 907.3 eV, and 916.4 eV for Ce3d_3/2_. These peaks can be identified in the Ce3d spectrum, characteristic of Ce^4+^ states [41,42,43,44]. The signal at 916.4 eV is particularly indicative of Ce^4+^ and is notably absent in Ce^3+^ [41,42,43,44,45,46]. This oxidation state prevails in films deposited at nitrogen fluxes of 30.0 sccm and 40.0 sccm, as depicted in Figure 8a.

Conversely, two pairs of 3d_5/2_–3d_3/2_ spin–orbit doublets characteristic of Ce^3+^ states at 880.8 eV, 885.3 eV, and 899 eV, 903.5 eV, respectively, emerge in the Ce3d spectrum of N15 and N25 flows (Figure 6 and Figure 8a). These states originate from distinct Ce4f-level occupancies in the final state and have been extensively discussed [46].

In films deposited with nitrogen flows of 20.0 sccm and 35.0 sccm, the presence of Ce^3+^ alongside Ce^4+^ on the film’s surface is evident. This is discernible by the observation of a shoulder around 881 eV, corresponding to the Ce^3+^ 3d_5/2_ 4f^1^ 3d^9^ state, situated between the Ce^4+^ peaks at 882 eV. These peaks correspond to a mixture of final states, specifically Ce^4+^ 3d_5/2_ 4f^2^ 3d^9^.

In the spectra depicting Ce, Zr, N_2_, and O_2_ for the five nitrogen fluxes utilized in this study (Figure 8), films deposited with nitrogen fluxes of N20 and N35 sccm exhibit characteristic 3d band-like spectra, providing evidence for the coexistence of Ce(III) and Ce(IV) oxidation states. For the spectral deconvolution process, the interpretation is well established in Figure 6. The first binding energy at 881 eV has been attributed to Ce^3+^, while the second at 882 eV is assigned to Ce^4+^, encompassing those corresponding to the previously noted three doublets [24,35,39,42,44]. The appended table to the right of Ce(III) and Ce(IV) identifies each of the binding energies for the Ce3d oxidation states. The peak at 816.4 eV, as indicated above, confirms the Ce^4+^ oxidation state. In Figure 8a, it is observed that films deposited with nitrogen flows of 15.0 sccm and 25.0 sccm predominantly exhibit Ce(III), whereas for films deposited with nitrogen flows of 20.0 sccm and 35.0 sccm, a mixture of Ce(III) and Ce(IV) is obtained. Finally, for nitrogen fluxes of 30.0 sccm and 40.0 sccm, the predominant species is Ce(IV).

The Zr3d spectra are characterized by the Zr3d_5/2_ and Zr3d_3/2_ doublets due to spin–orbit coupling as illustrated in Figure 8c. Additionally, the binding energies for Zr3p are presented. The predominant features in the Zr3d and Zr3p spectra are described below: at 181.6 eV for Zr3d_5/2_ and 184.0 eV for Zr3d_3/2_, with an integrated intensity of the Zr3d_3/2_ peak relative to that of the Zr3d_5/2_ peak equal to the spin–orbit multiplicity of 2/3 characteristic in ZrO_2_. The split width of the spin–orbit (3d_3/2_–3d_5/2_) is 2.3 eV, consistent across all deposit conditions. The shift of 0.1 eV towards lower binding energies for Zr3d would be associated with the substitution of oxygen for nitrogen in the crystalline structure of ZrCO_x_N_y_. For Zr3p, at 332.9 eV, the binding energy is attributed to spin–orbit coupling Zr3p_3/2_, and at 346.8 eV, to Zr3p_1/2_. The split width of the spin–orbit (3p_3/2_–3d_1/2_) is 13.8 eV with a spin–orbit multiplicity of 2/3, consistent across all deposit conditions [20,47,48].

The deconvolution results for the N1s and O1s spectra are presented in Figure 7. The O1s spectra exhibit asymmetry and can be curve-fitted with two peaks: a main peak at 531.4 ± 0.2 eV is attributed to ZrCeO_x_N_y_ and a peak at 530.0 eV is attributed to metal oxides: CeO_2_, Ce_2_O_3_, and/or ZrO_2_ [24,25,45,46]. No significant changes in oxygen binding energy are observed for the different deposition conditions. For N1s, the predominant binding energy is 398.0 eV, and another peak appears at 399 eV, both associated with oxynitride. The binding energy of 404 eV represents dinitrogen species adsorbed in the solid. The sample ZrCeO_x_N_y_ N25 exhibits an additional binding energy at 397.0 eV, which could indicate incipient nitride formation or further substitution of nitrogen for oxygen in this film. However, Ce3d, Zr3d, and Zr3p show no evidence of nitride formation [49,50].

For N20 and N40, there is a shift in Zr3d to 182.2 eV which would indicate a chemical environment with oxygen predominance around Zr. These two samples exhibit the lowest passivation zone in the corrosion tests.

#### 3.2.2. Chemical Structure

Cerium and zirconium were selected as targets because their oxides are known to form a stable solid solution [23,38] and crystallize in the same cubic structure, with a difference in lattice constants of 6.3% for CeO_2_–ZrO_2_ and 9.3% for Ce_2_O_3_–ZrO_2_. Figure 9 shows the diffractogram for the thin films deposited with increasing nitrogen fluxes from 15.0 to 40.0 sccm. To differentiate the crystallographic planes of the sample from those of the substrate, XRD of the AISI 316L used as substrate and a CeO_2_ sample are included.

The diffractograms provide evidence of crystallographic planes for cerium (IV) oxide (JCPDS 00-043-1002) and diffraction at 2θ = 65.0 and 82.0, absent in CeO_2_, which could be related to the formation of CeO_x_N_y_ [49]. Ce_2_O_3_ and ZrO_2_ are amorphous. Similar results have been reported for CSZT pH sensors and cerium oxide thin films deposited on Al 2024 alloy by reactive magnetron sputtering [24,48]. As the Pauling ionic radius of Zr^4+^ (0.084 nm) is smaller than that of Ce^4+^ (0.097 nm) and Ce^3+^ (0.104 nm), Zr^4+^ could gradually move into the CeO_2_ lattice to form a solid solution due to the same valence and similar ionic radius size, with no change in its interplanar distance and thus no change in the angle for the diffracting planes of CeO_2_.

In the enlargement of Figure 9b and Table 5, a shift towards the lower angle is observed in three different zones of the diffractogram for the Miller index planes (111), (200), and (220), for the samples deposited with nitrogen flow 15.0 sccm and 25.0 sccm, which would be associated with a greater substitution of oxygen for nitrogen. The other diffraction planes of the coating and substrate show no displacement. These two samples were characterized by having the highest chemical composition of nitrogen, as presented in Section 3.2.1.

The chemical composition of the film showed evidence of cerium, zirconium, nitrogen, and oxygen; therefore, the absence of zirconium crystalline phases and the lower chemical composition of zirconium suggest that it is found as a substituent in the crystal structure of CeO_x_N_y_ and hence only the CeO_2_ crystalline phase is identified. On the other hand, the Gibbs free energy of CeO_2_ (ΔG = −854.27 KJ/mol) is a thermodynamically more stable state than Ce_2_O_3_ (ΔG = −715.56 KJ/mol), favoring the formation of CeO_2_. Similar results have been reported by Wang Qi et al. during the formation of cerium zirconium oxide [38].

#### 3.2.3. Corrosion Test

In general, all ZrCeO_x_N_y_ films deposited with different nitrogen fluxes show higher corrosion resistance than the substrate. The corrosion current density (J_corr_) and corrosion rate (V_corr_) are lower by two orders of magnitude than the substrate. Meanwhile, the polarization resistance (R_p_) is higher by two orders of magnitude. Except for the sample deposited with a nitrogen flow of 20.0 sccm, for all samples, the corrosion potential (E_corr_), pitting nucleation potential (E_np_), and passivation zone (PZ) are higher than stainless steel. The film deposited with 15.0 sccm nitrogen flow has the largest passivation zone (Figure 10, Table 6). As an example, Figure 10b,c show the reproducibility of the corrosion curves for two random samples.

The chemical composition of nitrogen in the film and the oxidation state of cerium determine the corrosion resistance of the ZrCeO_x_N_y_ system. The corrosion constants together show that for the two samples with the highest percentage of nitrogen (approximately 5%), ZrCeO_x_N_y_ N15 and ZrCeO_x_N_y_ N25, the highest corrosion resistance is obtained. These two samples are also characterized by a predominance of the Ce^3+^ oxidation state in their chemical composition, while in those where the cerium is Ce^4+^, the corrosion resistance is lower compared to Ce^3+^. Similar behavior has been reported by different authors for alloys dosed with Ce(III). The authors report that the presence of Ce^3+^ is directly related to the increase in the corrosion resistance of Al–Co–Ce and AA7075 alloys, with the protective character of Ce(III) being higher than that of Ce(IV), which is the most reactive species [35,37,51]. In the Al–Co–Ce amorphous alloys, the passivating character of Ce^3+^ is due to its semiconductor behavior. It was observed that the presence of Ce_2_O_3_ increases the corrosion resistance more than CeO_2_ and the effect of Ce content can also be related to the oxygen content in the film, thus passivating the galvanized steel substrates by the formation of insoluble oxides and hydroxides. As already explained in Section 3.1, this result suggests that Ce^4+^ changes to Ce^3+^, due to the appearance of oxygen vacancies generated during film growth by sputtering.

Ershov et al. [52] modified the plasma conditions by varying the percentage of oxygen to deposit cerium oxide films. Concerning corrosion resistance, they observed that, as a function of oxygen percentage, the films can delaminate, exhibiting lower corrosion resistance than the substrate. However, films deposited with high oxygen fluxes are characterized by lower corrosion current densities than the substrate but maintain the pitting nucleation potential. In contrast, the ZrCeO_x_N_y_ films in the present work not only decrease the corrosion current density concerning the substrate by two orders of magnitude but also increase the pitting nucleation potential and, consequently, the passivation zone.

Ce^3+^ is more capacitive than Ce^4+^ because during the corrosion test, in the first step, Ce^3+^ is oxidized to Ce^4+^ without affecting the film morphology, since the ionic radius of Ce^3+^ is lower than that of Ce^4+^ [ionic radius of Ce^4+^ (0.87 Å) Ce^3+^ ion (1.10 Å)], as well as the unit cell volume [volume CeO_2_ = 158.5 × 10^6^ pm^3^, Ce_2_O_3_ = 175.05 × 10^6^ pm^3^]. Therefore, no cracks are generated in the film, leading to loss of film–substrate adhesion.

Moreover, the morphology of the ZrCeOxNy films deposited by RF sputtering shows dense films with thicknesses ranging from 414 to 615 nm, as seen in the cross-sectional view in Figure 11 and Table 7. In contrast, films deposited by co-sputtering are less dense, delaminate, and their thickness does not exceed 250 nm under the same deposition conditions. The films deposited by co-sputtering exhibit a layered growth with a granular surface, as observed in the cross-sectional and surface views in Figure 12a, whereas RF sputtering results in columnar growth and approximately 50% greater thickness. The corrosion results indicate that the films with the largest passivation zone are the thickest ones (see Table 6 and Table 7).

To assess the damage caused to the films after the corrosion test, scanning electron microscopy was employed to evaluate changes in the surface morphology of the coating, and X-ray diffraction was utilized to determine alterations in the crystalline structure. Figure 13 illustrates the morphology of the ZrCeO_x_N_y_ films before and after the corrosion test. The letter ‘N’ followed by a number indicates the nitrogen flow rate used during the deposition; thus, N30 corresponds to the film deposited with a nitrogen flow rate of 30.0 sccm. The remaining deposition conditions are kept constant for all samples. Figure 13a presents the morphology for the ZrCeO_x_N_y_ films deposited with different nitrogen fluxes, showing that all films are dense and homogeneous, with very few surface imperfections. Figure 12b displays the surface morphology after corrosion.

ZrCeO_x_N_y_ deposited with a 15.0 sccm nitrogen flow shows very few imperfections before and after the corrosion test. This evidence, associated with the corrosion test, would explain why it has a wide passivation zone (PZ = 1.23 V), compared to the substrate and films deposited under different conditions.

Films deposited at 20.0 sccm, 30.0 sccm, and 35.0 sccm nitrogen fluxes (N20, N30, and N35, respectively) exhibit delamination of the film until it reaches the substrate in the corrosion zone. At N40, there is delamination, but the substrate is not reached. The surface of the ZrCeO_x_N_y_ film deposited with ϕN_2_ = 35 sccm (Figure 11a) shows nucleation zones where the coating grows on the surface and micropore-like surface defects. After being subjected to corrosion tests (Figure 13b), extensive surface damage is observed in the corrosion zone, where delamination of the film and penetration of the electrolyte into the substrate are evident. On the contrary, in Figure 11a, it is observed that the ZrCeO_x_N_y_ film deposited with ϕN_2_ = 40.0 sccm is dense, homogeneous, and exhibits few imperfections. After the corrosion test in Figure 13b, pitting is observed in the corrosion zone, but the coating remains adhered to the substrate.

The XRD results in Figure 14 depict the sample in orange before the corrosion test and in green after the potentiodynamic polarization test. The outcomes indicate that following the corrosion test, the ZrCeO_x_N_y_ N15 film exhibits in-plane growth (200), while both ZrCeO_x_N_y_ N20 and ZrCeO_x_N_y_ display in-plane growth (111). Generally, there is no evident damage to the crystalline structure of the films, except for the sample deposited with a nitrogen flow of 40.0 sccm, where the CeO_2_ peaks appear broad after corrosion, implying a transformation to an amorphous phase (refer to Figure 13 N40-b). In contrast, the film deposited with a nitrogen flux of N35 does not exhibit structural damage, although it does reveal surface morphology damage after the corrosion test, as illustrated in Figure 13 N35-b.

ZrCeO_x_N_y_ films are characterized by a homogeneous composition with evidence of nitrogen presence. Regarding their crystalline structure, films deposited by co-sputtering are strongly textured, with preferential growth on the (220) plane for the film grown with a nitrogen flow of 15.0 sccm, and on the (200) plane for films deposited with a nitrogen flow of 20.0 sccm. These results are reproducible on glass. Meanwhile, films deposited by the RF sputtering technique are polycrystalline but do not exhibit preferential growth on any particular plane. The morphology of films deposited by co-sputtering showed low adhesion to the substrate, but this does not detract from their corrosion resistance, which is superior by one or two orders of magnitude compared to the substrate. By the RF sputtering technique, ZrCeO_x_N_y_ films do not exhibit delamination from the substrate, are dense with columnar growth, their corrosion resistance is superior to that of the steel for all nitrogen flows studied in this work, and after being subjected to corrosion tests, the coating loss is minimal in the corrosion zone and the crystalline structure is preserved.

Quiao and Merguel have reported that the plasma is more concentrated in the erosion track, which generates a difference in the morphology, growth type, and crystalline structure of the films, as the plasma in an RF discharge is more extended [53]. On the other hand, Timoshnev et al. found that the concentration of ionic species in the plasma is determined by the film deposition method [54]. The good characteristics of the ZrCeO_x_N_y_ films deposited by RF sputtering are directly related to the direct discharge of the sputtered atoms from the target to the substrate, which increases the plasma density and its confinement by magnetic field lines, producing a dense, stable coating with excellent corrosion resistance.

While the co-sputtering method allows for the variation of the film’s chemical composition without breaking the vacuum, there is also the possibility of introducing multiple targets into the vacuum chamber simultaneously. This enables the creation of complex films with different materials. The target configuration used in this work allows for the deposition of a film composed of four chemical elements, with a dense and homogeneous morphology and high corrosion resistance, and without vacuum loss, since all four elements are deposited simultaneously. The arrangement of the target oriented parallel to the substrates facilitates the mean free path for atoms to travel from the target to the substrate. On the other hand, in the co-sputtering technique, to achieve a more efficient and homogeneous deposition, the position of the targets is critical due to their angled orientation relative to the substrate, which prevents a direct focus towards it. This likely generates substrate-coating tensile stresses, leading to the delamination of the coating.

The application of unbalanced magnetic fields ensures that the particles involved in the plasma collide multiple times, generating a higher-density plasma. Although this may affect the mean free path, it ensures that the ions of the materials present in the deposit move along the open magnetic field lines and can reach the substrate with sufficiently high energy to produce coatings with high adhesion.

## 4. Conclusions

Oxynitrides of certain transition metals (TiO_x_N_y_, HfO_x_N_y_, ZrOxN_y_) have demonstrated significantly higher corrosion resistance and biocompatibility compared to their oxides, owing to the combined properties of oxide and nitride within the same crystalline structure. While the oxides of these metals exhibit high corrosion resistance, oxynitrides react in corrosive environments to form oxides, thereby enhancing their resistance due to the chemical inertness of the initial reaction product (oxide). Research into the biocompatibility of transition metal oxynitrides represents a burgeoning field aimed at producing materials that do not release ions causing cytotoxicity in biological environments.

The excellent inert and non-toxic properties of these materials stem from their corrosion resistance, which benefits surgical-grade steel commonly used in orthopedic applications. The present study demonstrates that the synthesized coating based on cerium zirconium oxynitride reduces the corrosion rate of AISI 316L in biological media by two orders of magnitude, thereby extending its operational lifespan without degradation.

From the reaction of two transition metals with high chemical inertness (Zr and Ce) in a reactive nitrogen–oxygen atmosphere, ZrCeO_x_N_y_ coatings were obtained with a dense morphology, high homogeneity, and corrosion resistance superior to that of surgical-grade stainless steel. Their chemical stability and good adhesion to the substrate are reflected after being subjected to severe corrosion conditions, where they maintain their morphology and crystalline structure. ZrCeO_x_N_y_ thin films, synthesized through both co-sputtering and RF sputtering techniques, emerge as highly promising coatings for surgical-grade AISI 316L stainless steel, particularly in the context of osteosynthesis applications. The superior corrosion resistance displayed by these films, compared to that of the substrate, underscores their potential for enhancing the durability and longevity of orthopedic implants.

Morphological assessment of RF-sputtered films reveals a dense and homogeneous structure, qualities crucial for ensuring robust adhesion and effective protection against corrosion. However, co-sputtered films tend to delaminate from the substrate.

Chemical composition analyses reveal a critical factor in corrosion resistance: films containing cerium in oxidation state (III), or a balanced combination of Ce(III) and Ce(IV), demonstrate enhanced corrosion resistance compared to those predominantly featuring a Ce(IV) oxidation state. This finding not only contributes to the understanding of the protective mechanisms of ZrCeO_x_N_y_ films but also provides guidance for optimizing their composition to maximize performance.

The crystalline structure of ZrCeO_x_N_y_ films, predominantly characterized by the cubic CeO_2_ phase, remains remarkably stable even after exposure to corrosive environments. This stability, coupled with the films’ corrosion resistance (two orders of magnitude higher than AISI 316L), positions ZrCeO_x_N_y_ as a robust and reliable candidate for coating surgical-grade stainless steel, potentially mitigating issues related to corrosion and ensuring the longevity of orthopedic implants.

In summary, the assessment of ZrCeO_x_N_y_ thin films from different perspectives, encompassing morphology, chemical composition, and crystalline structure, reinforces their candidacy as protective coatings in orthopedic applications. The exploration of different deposition techniques and the understanding of the influence of cerium oxidation states contribute valuable knowledge to the ongoing quest for corrosion-resistant materials in the biomedical field.

## Figures and Tables

**Figure 1 materials-17-03142-f001:**
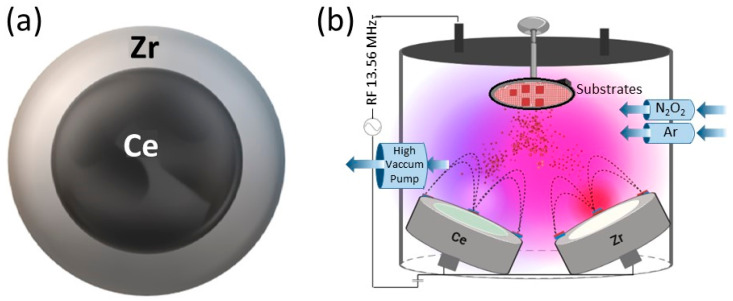
Schematic representation of sputtering system. (**a**) Target configuration used in RF sputtering. Zirconium ring delimiting the cerium target. (**b**) Schematic representation of RF co-sputtering.

**Figure 2 materials-17-03142-f002:**
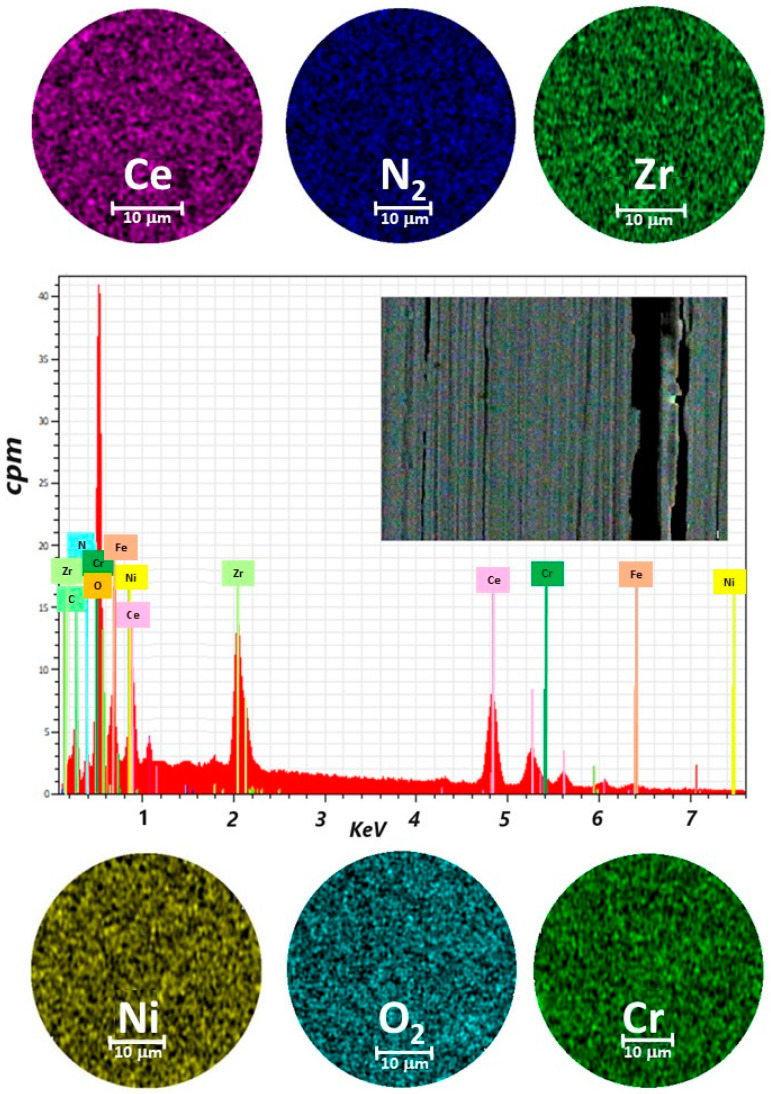
Elemental mapping image for Ce, Zr, N_2_, O_2_, Cr, Ni, and EDX spectrum of the selected area of coatings of ZrCeO_x_N_y_ deposited on stainless steel 316L via co-sputtering technique.

**Figure 3 materials-17-03142-f003:**
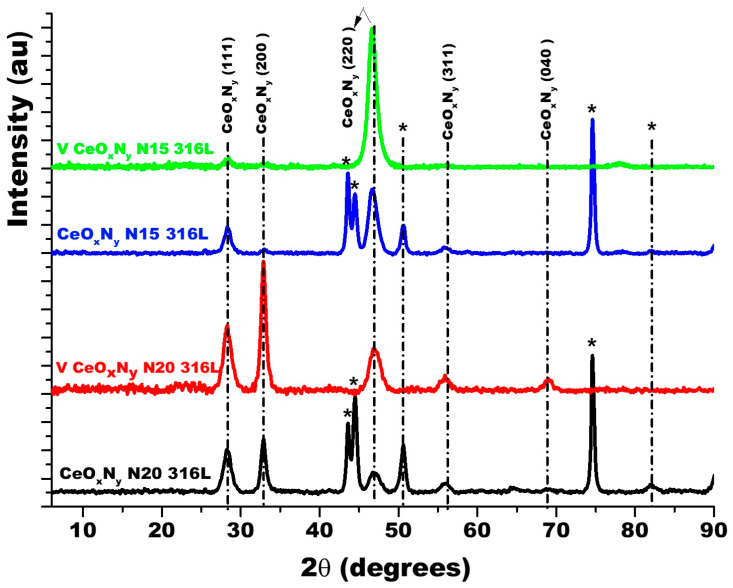
XRD of ZrCeO_x_N_y_ coatings deposited with different nitrogen fluxes on 316L stainless steels by the co-sputtering method. Additionally, the XRD of the substrate and CeO_2_ powder is presented. Phases of AISI 316L are designated as (*).

**Figure 4 materials-17-03142-f004:**
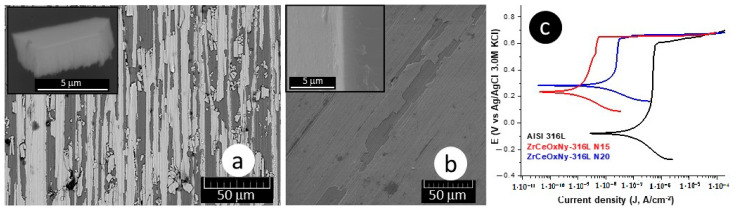
Morphology: (**a**) ZrCeO_x_N_y_-316L N15, (**b**) ZrCeO_x_N_y_-316L N20, and (**c**) polarization curves for ZrCeO_x_N_y_ thin films obtained via co-sputtering.

**Figure 5 materials-17-03142-f005:**
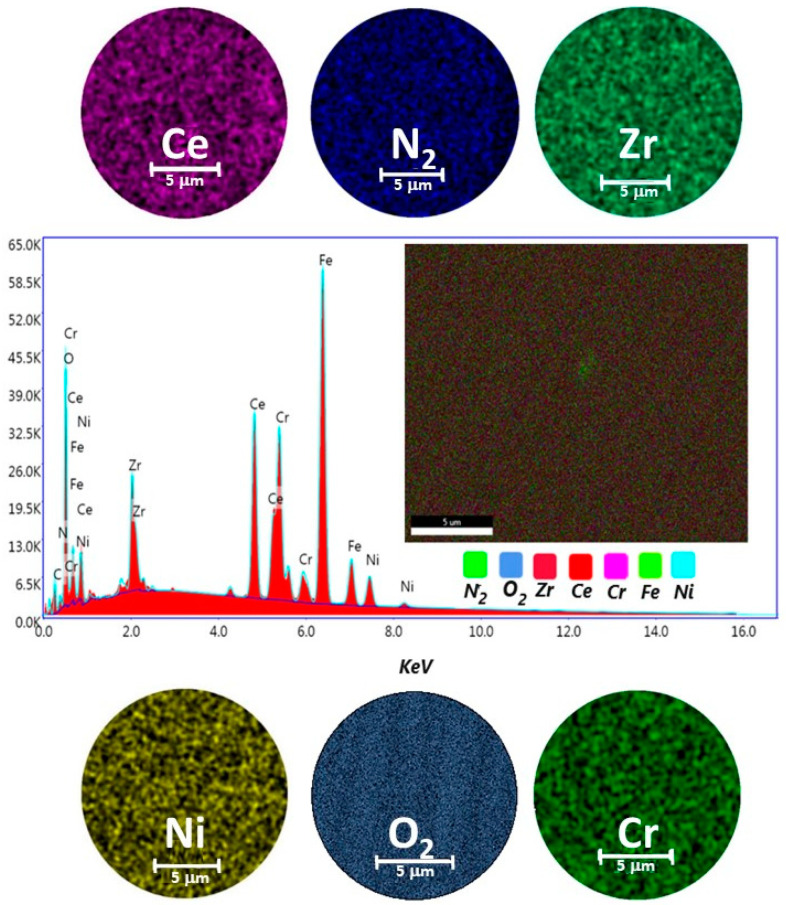
Elemental mapping image for Ce, Zr, N_2_, O_2_, Cr, Ni, and EDX spectrum of the selected area of ZrCeO_x_N_y_ coatings deposited on stainless steel 316L via the RF sputtering technique.

**Figure 6 materials-17-03142-f006:**
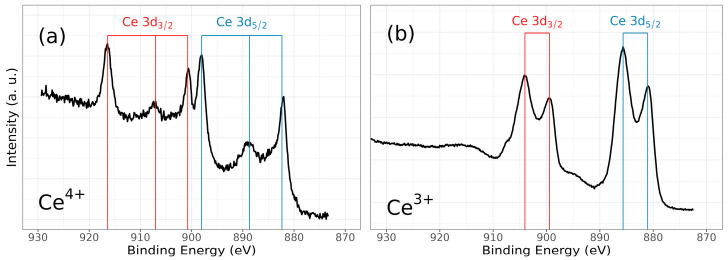
High-resolution XPS spectra of ZrCeO_x_N_y_, for (**a**) Ce3d (IV) and (**b**) Ce3d (III), and after cleaning with Ar^+^ for 5 min.

**Figure 7 materials-17-03142-f007:**
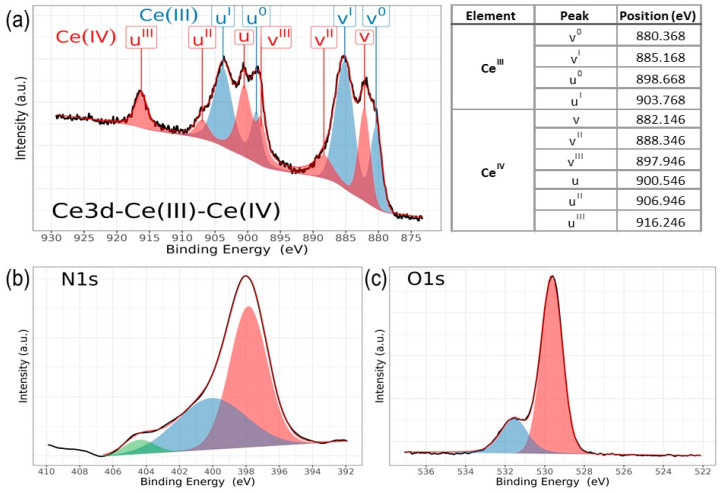
High-resolution XPS spectra of ZrCeO_x_N_y_, for (**a**) a mix of Ce3d (III) and Ce3d (IV), (**b**) N1s, and (**c**) O1s, deposited under different nitrogen flow conditions and after cleaning with Ar^+^ for 5 min.

**Figure 8 materials-17-03142-f008:**
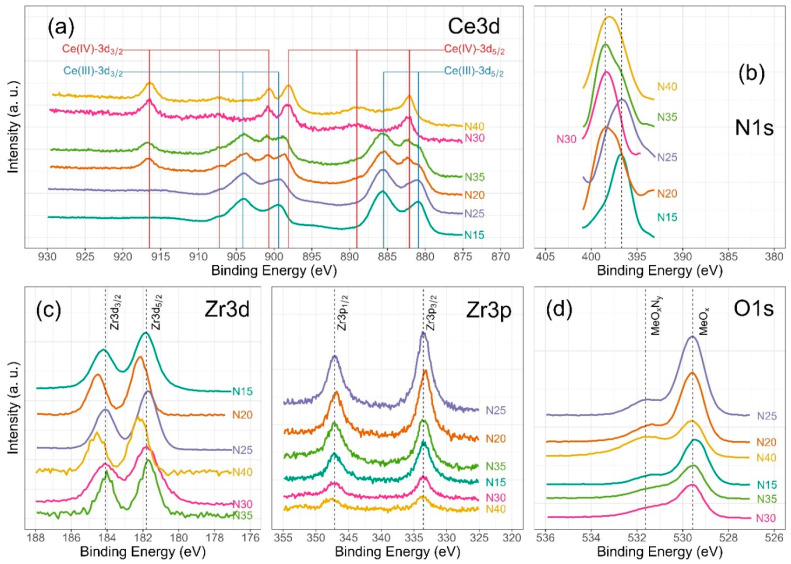
High-resolution XPS spectra of ZrCeO_x_N_y_, for (**a**) Ce3d, (**b**) N1s, (**c**) Zr3d and Zr3p, and (**d**) O1s deposited under different nitrogen flow conditions and after cleaning with Ar^+^ for 5 min.

**Figure 9 materials-17-03142-f009:**
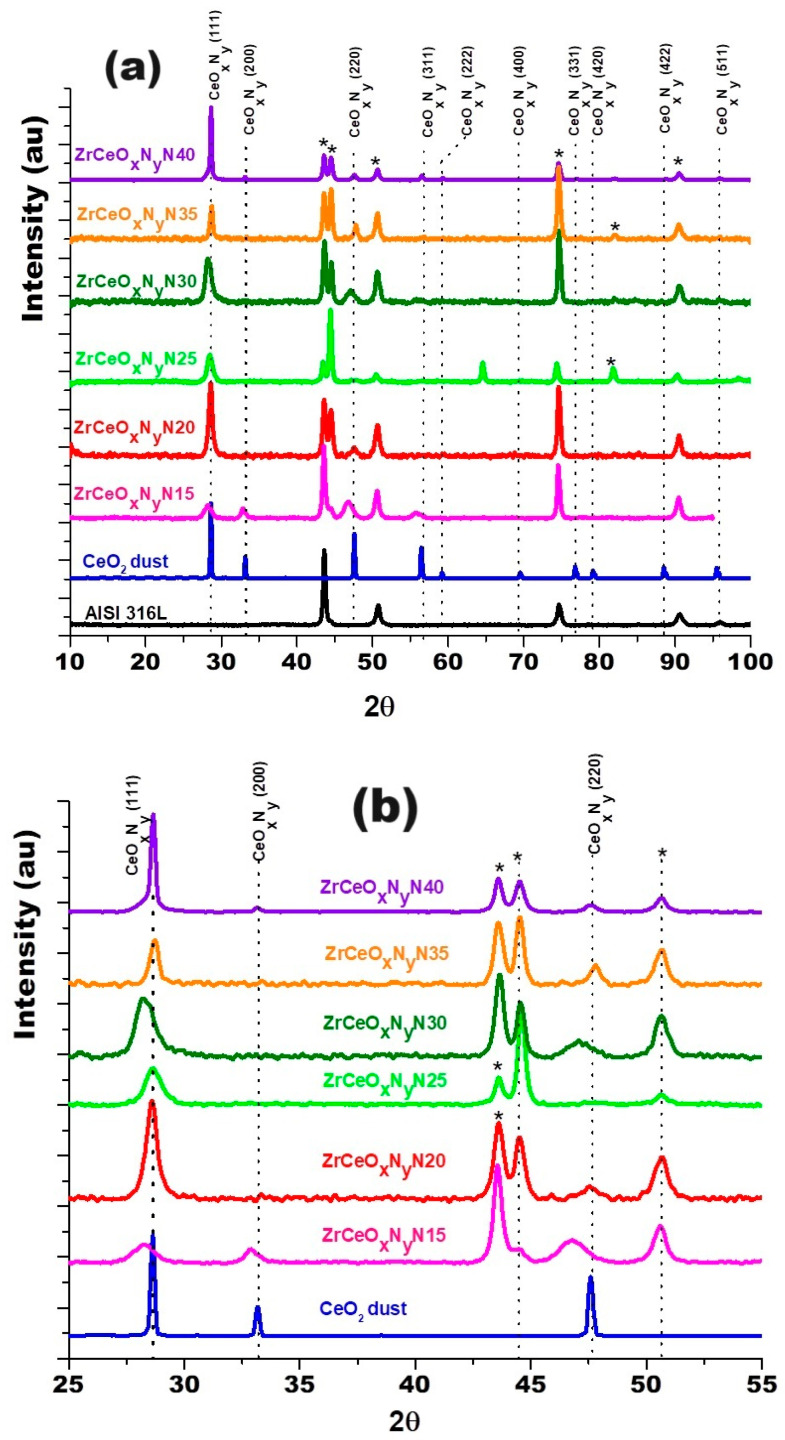
XRD of ZrCeO_x_N_y_ coatings deposited with different nitrogen fluxes on 316L stainless steels by the RF sputtering method. Additionally, the XRD of the substrate and CeO_2_ powder is presented. Phases of AISI 316L are designated as (*).

**Figure 10 materials-17-03142-f010:**
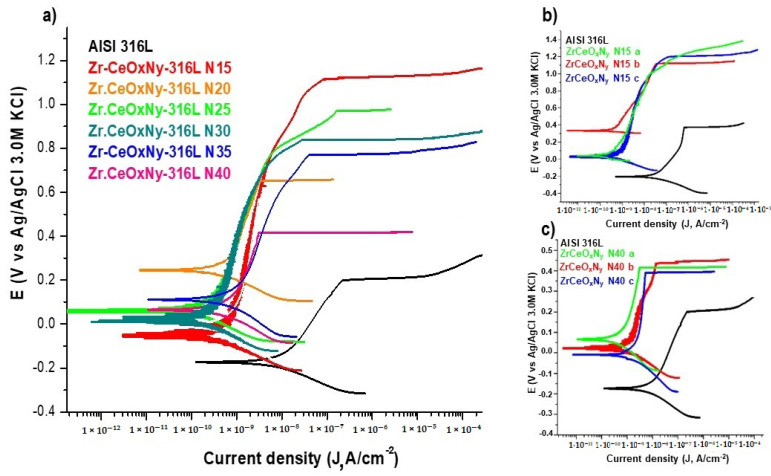
Corrosion tests of samples tested in Hank’s solution. (**a**) Potentiodynamic polarization for ZrCeO_x_N_y_-316L deposited with different nitrogen flow, (**b**) triplicate for ZrCeO_x_N_y_ N15-316L, and (**c**) triplicate for ZrCeO_x_N_y_ N40-316L. Bare AISI-316L is introduced as a reference.

**Figure 11 materials-17-03142-f011:**
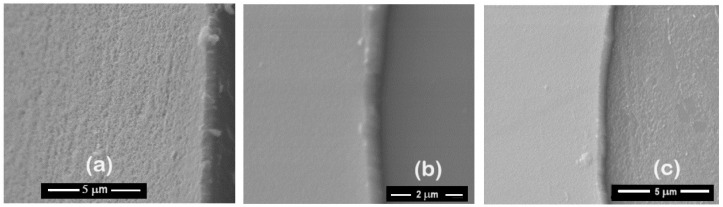
Morphology of ZrCeO_x_N_y_ coatings deposited by RF sputtering with different nitrogen fluxes: ϕN_2_ = 15 sccm (**a**), 25 sccm (**b**), and 30 sccm (**c**).

**Figure 12 materials-17-03142-f012:**
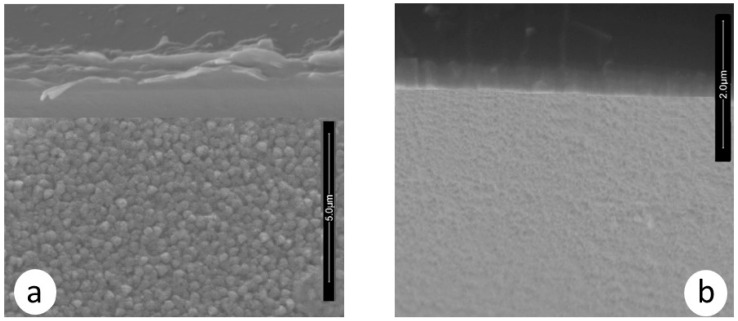
Morphology in cross-section and surface section for ZrCeO_x_N_y_ coatings deposited via co-sputtering (**a**) vs. RF sputtering (**b**).

**Figure 13 materials-17-03142-f013:**
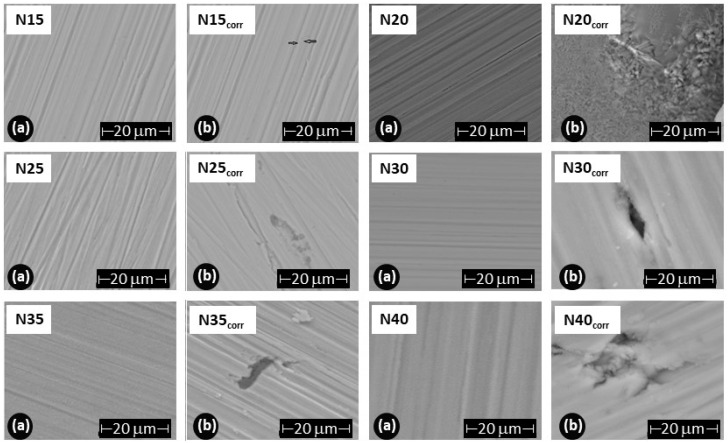
Morphology of ZrCeO_x_N_y_ coatings deposited by RF sputtering with different nitrogen fluxes: ϕN_2_ = 15, 20, 25, 30, 35, and 40 sccm. (**a**) Before corrosion, (**b**) after corrosion.

**Figure 14 materials-17-03142-f014:**
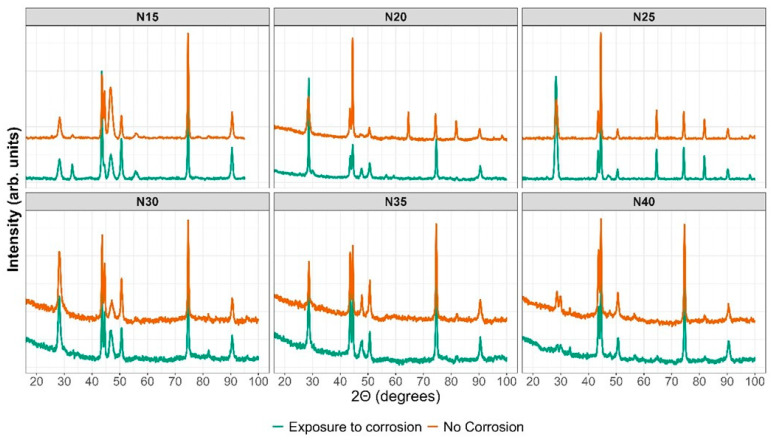
XRD before and after corrosion.

**Table 1 materials-17-03142-t001:** Conditions tested in each method.

Coatings	Power (W)	T (°C)	ϕAr (sccm)	ϕO_2_ (sccm)	ϕN_2_ (sccm)
ZrCeO_x_N_y_ RF sputtering	150, 200, 250, 300, 350	200, 220, 300	20.0	1.00	15.0–40.0
				1.50	
				2.00	
ZrCeO_x_N_y_ Co-sputtering	100, 150, 200	150, 200, 200	20.0	1.00	15.0–40.0
				1.50	
				2.00	

**Table 2 materials-17-03142-t002:** Chemical composition of Hank´s solution used in corrosion test.

Compound	NaCl	CaCl_2_·2H_2_O	KCl	MgCl_2_·6H_2_O	NaH_2_PO_4_	KH_2_PO_4_	MgSO_4_·7H_2_O	NaHCO_3_	C_6_H_12_O_6_
Concentration (g/L)	8.00	0.140	0.400	0.100	6.00 × 10^−2^	6.00 × 10^−2^	6.00 × 10^−2^	0.450	1.00

**Table 3 materials-17-03142-t003:** Average values of the parameters of the corrosion test samples in Hank’s solution using the co-sputtering technique.

Sample	ϕN_2_ (sccm)	J_corr_ (A/cm^2^)	E_corr_ (V)	R_p_ (Ohm/cm^2^)	V_corr_ (mmpy)	E_np_ (V)
AISI 316L		6.75 × 10^−8^	−7.80 × 10^−2^	3.85 × 10^5^	7.84 × 10^−4^	5.87 × 10^−2^
ZrCeO_x_N_y_ N15	15.0	8.16 × 10^−10^	2.34 × 10^−1^	3.19 × 10^7^	9.49 × 10^−6^	6.49 × 10^−1^
ZrCeO_x_N_y_ N20	20.0	8.38 × 10^−9^	2.83 × 10^−1^	3.10 × 10^6^	9.74 × 10^−5^	6.44 × 10^−1^

**Table 4 materials-17-03142-t004:** Elemental composition for ZrCeO_x_N_y_ thin films deposited by RF sputtering. (a) chemical composition for the coating elements, inclusive of the substrate alloying elements (Fe, Cr, and Ni). (b) Only the coating chemical elements Zr, Ce, and N_2_ have been normalized to 100%.

(a)						
Element	Weight %					
	Zr-CeO_x_N_y_ N15	Zr-CeO_x_N_y_ N20	Zr-CeO_x_N_y_ N25	Zr-CeO_x_N_y_ N30	Zr-CeO_x_N_y_ N35	Zr-CeO_x_N_y_ N40
N (K)	1.12	0.802	1.17	0.820	0.65	0.601
O (K)	11.5	11.6	10.6	8.05	10.4	9.30
Zr (L)	5.87	7.86	4.28	4.24	6.00	3.76
Ce (L)	15.9	24.8	17.7	16.5	13.6	16.9
Cr (K)	11.8	17.3	14.6	7.6	12.6	12.1
Fe (K)	38.2	25.3	40.6	54.2	41.2	42.7
Ni (K)	8.96	4.55	5.66	4.65	9.06	5.95
**(b)**						
**Element**	**Weight %**					
	**Zr-CeO_x_N_y_** **N15**	**Zr-CeO_x_N_y_** **N20**	**Zr-CeO_x_N_y_** **N25**	**Zr-CeO_x_N_y_** **N30**	**Zr-CeO_x_N_y_** **N35**	**Zr-CeO_x_N_y_** **N40**
N (K)	4.10	2.90	5.05	3.80	3.21	2.83
Zr (K)	25.5	26.1	18.5	19.6	29.7	17.7
Ce (L)	70.4	71.0	76.5	76.6	67.1	79.5

**Table 5 materials-17-03142-t005:** Structural changes in ZrCeO_x_N_y_ by the RF sputtering method.

2θ	h k l	CeO_2_	ZrCeO_x_N_y_	Δθ
1	1 1 1	28.8	27.7	1.1
2	2 0 0	33.3	32.8	0.45
3	2 2 0	47.6	46.8	0.83

**Table 6 materials-17-03142-t006:** Average values of the parameters of the corrosion test samples in Hank’s solution via the co-sputtering technique.

Sample	ϕN_2_ sccm	J_corr_ nA/cm^2^	E_corr_ V	R_p_ MΏ/cm^2^	V_corr_ mmpy10^−6^	E_np_ V	PZx 10^−1^
AISI 316L	NA	67.5 ± 6.8	−0.174 ± 4.3 × 10^−2^	0.385 ± 0.11	784 ± 68	0.204 ± 3.8 × 10^−2^	3.78
ZrCeO_x_N_y_ N15	15.0	0.412 ± 0.12	−9.60 × 10^−2^ ± 8 × 10^−3^	63.1 ± 17	1.89 ± 0.37	1.13 ± 0.27	12.3
ZrCeO_x_N_y_ N20	20.0	0.453 ± 6.8 × 10^−2^	0.225 ± 4.0 × 10^−2^	57.4 ± 6.5	2.07 ± 0.34	0.644 ± 0.16	4.19
ZrCeO_x_N_y_ N25	25.0	0.210 ± 2.8 × 10^−2^	2.05 × 10^−2^ ± 9.4 × 10^−3^	129 ± 15	2.35 ± 0.34	0.777 ± 0.17	7.32
ZrCeO_x_N_y_ N30	30.0	0.206 ± 0.17 × 10^−1^	0.266 ± 2.0 × 10^−2^	12.6 ± 3.4	2.37 ± 0.20	0.790 ± 0.22	7.87
ZrCeO_x_N_y_ N35	35.0	0.539 ± 0.11	0.306 ± 6.0 × 10^−2^	48.3 ± 12	6.26 ± 1.6	0.960 ± 0.17	6.54
ZrCeO_x_N_y_ N40	40.0	0.450 ± 0.12 × 10^−3^	7.55 × 10^−2^ ± 5.1 × 10^−3^	57.8 ± 4.0	2.06 ± 1.5 × 10^−1^	0.409 ± 0.69 × 10^−1^	3.34

**Table 7 materials-17-03142-t007:** Thickness for ZrCeO_x_N_y_ thin films deposited by RF sputtering with different nitrogen fluxes.

ZrCeO_x_N_y_	N15	N20	N25	N30	N35	N40
**Thickness (nm)**	615 ± 8.8	464 ± 16	526 ± 7.5	540 ± 17	506 ± 27	414 ± 6.2

## Data Availability

The raw data supporting the conclusions of this article will be made available by the authors on request.

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
