# Peer review of "Thin Layers of Cerium Oxynitride Deposited via RF Sputtering"

_materials, 2024, doi:10.3390/ma17133142_

Round 1

Reviewer 1 Report

Comments and Suggestions for Authors

- Can you provide more details on the consistency and repeatability of your substrate preparation process, especially considering the variability in SiC grain sizes used for polishing?

- Why were the specific power levels (250 W for co-sputtering and 150 W for RF sputtering) chosen for your deposition processes? How do these parameters impact the film characteristics?

- What criteria were used to select the accelerating voltages for SEM imaging? Did you observe any differences in surface morphology at different voltages?

- How were the phases identified using XRD? Were there any challenges in differentiating between CeO2 and potential CeOxNy phases?

- How representative are the test conditions (using Hank's solution) of the actual service environment for these coatings?

- Can you elaborate on the procedure and assumptions used for the Tafel extrapolation method to determine Ecorr and Jcorr?

- How does the variation in nitrogen flux affect the mechanical properties of the films, such as hardness and adhesion strength?

- How consistent were your results across multiple samples and deposition runs? Can you provide statistical data to support the reproducibility of your findings?

- How do your results compare with similar studies in the literature, particularly those involving CeOxNy and Zr-based coatings?

Reviewer 2 Report

Comments and Suggestions for Authors

The article "Thin layers of cerium oxynitride deposited via RF sputtering" discusses the deposition and analysis of cerium zirconium oxynitride thin films on surgical-grade stainless steel. These films aim to enhance the corrosion resistance of the steel, particularly for biomedical applications such as osteosynthesis.

·       To further enhance the comprehensiveness of your study, I suggest including a comparative analysis between DC magnetron sputtering and RF sputtering techniques. This comparison will provide a clearer context for the advantages of RF magnetron sputtering used in your work.

see for example

1.       Comparison of radio-frequency and direct-current magnetron sputtered thin In2O3:Sn films,

Thin Solid Films, Volume 484, Issues 1–2, 2005, Pages 146-153, ISSN 0040-6090,

https://doi.org/10.1016/j.tsf.2005.02.006.

2.       Graphene oxide on magnetron sputtered silver thin films for SERS and metamaterial applications,

Applied Surface Science, Volume 427, Part B, 2018, Pages 927-933, ISSN 0169-4332, https://doi.org/10.1016/j.apsusc.2017.09.059.

3 Annealing Temperature Effect on the Physical Properties of NiO Thin Films Grown by DC Magnetron Sputtering. Adv. Mater. Interfaces 2024, 11, 2300815. https://doi.org/10.1002/admi.202300815

·       Given the potential application in biomedical implants, have you conducted or considered in vitro or in vivo biocompatibility tests for the ZrCeOxNy coatings?

·       The authors should discuss better the novelty of their work in comparison to previous research works.

·       What are the future directions or next steps in this research?The authors could discuss better this aspect in the conclusion sections.

Comments on the Quality of English Language

moderate revision 

Round 2

Reviewer 2 Report

Comments and Suggestions for Authors

I recommend the publication of this work